# Sex-biased regulatory changes in the placenta of native highlanders contribute to adaptive fetal development

Tian Yue[1,2†], Yongbo Guo[1,2†], Xuebin Qi[1,3,4†], Wangshan Zheng[1†], Hui Zhang[1,4†], Bin Wang[3], Kai Liu[1], Bin Zhou[1,2], Xuerui Zeng[1,2], Ouzhuluobu[3], Yaoxi He[1*], Bing Su[1,5*]

[1]State Key Laboratory of Genetic Resources and Evolution, Kunming Institute of Zoology, Chinese Academy of Sciences, Kunming, China; [2]Kunming College of Life Science, University of Chinese Academy of Sciences, Beijing, China; [3]Fukang Obstetrics, Gynecology and Children Branch Hospital, Tibetan Fukang Hospital, Kunming, China; [4]State Key Laboratory of Primate Biomedical Research, Institute of Primate Translational Medicine, Kunming University of Science and Technology, Kunming, China; [5]Center for Excellence in Animal Evolution and Genetics, Chinese Academy of Sciences, Kunming, China

*For correspondence:
heyaoxi@mail.kiz.ac.cn (YH);
sub@mail.kiz.ac.cn (BS)

†These authors contributed equally to this work

Competing interest: The authors declare that no competing interests exist.

**Abstract** Compared with lowlander migrants, native Tibetans have a higher reproductive success at high altitude though the underlying mechanism remains unclear. Here, we compared the transcriptome and histology of full-term placentas between native Tibetans and Han migrants. We found that the placental trophoblast shows the largest expression divergence between Tibetans and Han, and Tibetans show decreased immune response and endoplasmic reticulum stress. Remarkably, we detected a sex-biased expression divergence, where the male-infant placentas show a greater between-population difference than the female-infant placentas. The umbilical cord plays a key role in the sex-biased expression divergence, which is associated with the higher birth weight of the male newborns of Tibetans. We also identified adaptive histological changes in the male-infant placentas of Tibetans, including larger umbilical artery wall and umbilical artery intima and media, and fewer syncytial knots. These findings provide valuable insights into the sex-biased adaptation of human populations, with significant implications for medical and genetic studies of human reproduction.

## eLife assessment

This **fundamental** study reports differential expression of key genes in full-term placenta between Tibetans and Han Chinese at high elevations, which are more pronounced in the placenta of male fetus than in female fetus. The gene expression data were collected and analyzed using **solid** and validated methodology, although there is limited support for hypoxia-specific responses due to a lack of low-altitude samples. Several of the placental genes found in this study have been previously reported to show signatures of positive selection in Tibetans, pointing to a potential mechanism of how human populations adapt to high elevation by mitigating the negative effects of low oxygen on fetal growth. The work will be of interest to evolutionary and population geneticists as well as researchers working on human hypoxic response.

## Introduction

Hypobaric hypoxia at high altitude can restrict fetal growth, resulting in reduced neonatal birth weight (BW) and increased infant mortality (*Moore et al., 2001*). Due to long-term natural selection at high altitude, native highlanders such as Tibetans have acquired a higher reproductive success than the lowlander migrants (e.g. Han Chinese moving to high altitude), reflected by the higher newborn BW and the lower prenatal and postnatal mortality (*Beall et al., 2004*; *Moore et al., 2011*). As expected, Tibetans performed better than Han migrants during fetal development (*He et al., 2023*; *Julian et al., 2009*; *Moore, 2001*; *Moore et al., 2011*). Previous genomic studies have identified a group of genes potentially contributing to the genetic adaptation of Tibetans to high-altitude environments, and these studies were mostly focused on the blood and cardiopulmonary systems of adult Tibetans (*Beall et al., 2010*; *Bigham et al., 2010*; *Deng et al., 2019*; *Peng et al., 2017*; *Peng et al., 2011*; *Simonson et al., 2010*; *Song et al., 2020*; *Xiang et al., 2013*; *Xu et al., 2011*; *Yang et al., 2017*; *Yi et al., 2010*). Although a research reported the association of more pregnancies and a higher survival birth rate based on GWAS analysis of 1008 Nepali Tibetans, no adaptive genetic variant was identified (*Jeong et al., 2018*). Therefore, whether and how the Tibetan adaptive variants affect their fetal development and eventually improve the reproductive fitness are yet to be explored.

The placenta is a temporary organ that connects fetus to the mother's uterus during pregnancy, and it begins to develop around 6–7 days after gestation (*Knöfler et al., 2019*). As an important organ of maternal-fetal exchange of nutrient, oxygen, and waste, the placenta plays a critical role in fetal growth and development. Previous studies have reported that placental dysfunction can lead to premature birth, abnormal fetal growth, and defects of neurological development (*Guttmacher et al., 2014*). At the same time, the genetic study reported that placental transcription in utero plays a key role in determining postnatal body size (*Peng et al., 2018*), and placenta weight is positively correlated with BW, an important trait related to the postnatal survival rate of newborns (*Haeussner et al., 2013*). Therefore, gene expression profiling of placenta is essential to dissecting the molecular mechanism of the successful reproduction. Recently, a transcriptomic study comparing 91 placental samples (including 47 Tibetans giving birth at high altitudes) found that the genes related to autophagy and tricarboxylic acid cycle showed significant up-regulation in Tibetans. However, no lowlander migrants living at high altitude (>2500 m) were included in this study (*Wu et al., 2022*).

In addition, it is known that the intrauterine growth rate of the male fetus is faster than that of the female fetus, and this growth rate difference leads to a higher proportion of preeclampsia, premature delivery, and intrauterine growth restriction of the male fetus (*Clifton, 2010*). The between-sex difference of fetal development was explained by the higher transport efficiency, but weaker reserve capacity of the male fetal placenta (*Eriksson et al., 2010*). Consistently, we recently reported a male-biased pattern of BW seasonality in Tibetans and Han migrants living at high altitude. Compared to the female fetuses, the male fetuses are more sensitive to hypobaric hypoxia, reflected by their more serious BW reduction and relatively lower survival rates (*He et al., 2022*). However, whether the placenta plays a key role in the observed sex-biased sensitivity to hypoxia during fetal development at high altitude remains elusive.

In this study, we sampled 69 full-term placental tissues, including 35 native Tibetans and 34 Han migrants living at the same altitude (an elevation of 3650 m). Seven tissue layers of the placenta were dissected and analyzed in detail. With the use of RNA-seq, we generated a systematic map of placental transcriptomes of indigenous and migrant populations living at high altitude. Markedly, we found a male-biased transcriptomic divergence between Tibetans and Han migrants, and the gene expression pattern of the fetal umbilical cord (UC) may affect the BW of the male newborns. We also observed adaptive histological changes in the Tibetan male-infant placentas, providing further evidence on the sex-biased adaptation of fetal development.

## Results

### Gene expression profiles of placentas of native Tibetans and Han migrants

To understand the transcriptomic patterns of placentas of Tibetans and Han migrants at high altitude, we collected full-term (>37 weeks of gestation) placental samples from 35 Tibetans (16 male newborns and 19 female newborns) and 34 Han migrants (21 male newborns and 13 female newborns)

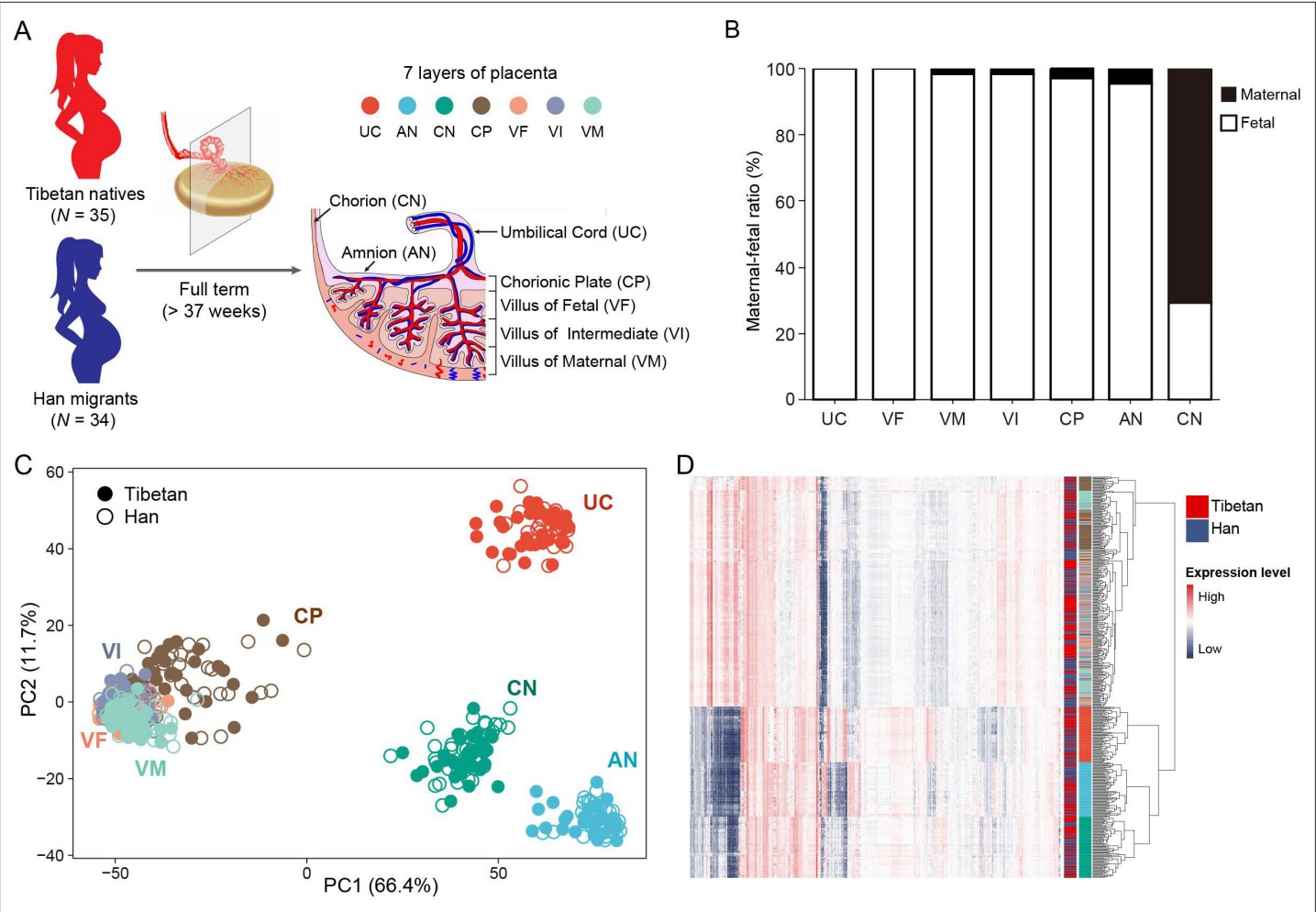

**Figure 1.** Sampling strategy and gene expression patterns of the placental layers of Tibetans and Han migrants at high altitude. (**A**) The strategy of sampling the full-term placentas. The placenta was dissected into seven layers, as shown from the fetal side to the maternal side, are umbilical cord (UC), amnion (AN), chorion (CN), chorionic plate (CP), villus of fetal (VF), villus of intermediate (VI), and villus of maternal (VM). The seven layers are labeled with seven different colors. (**B**) Analysis of the maternal-fetal origins of the placental layers. (**C**) The map of principal component analysis (PCA) showing the clustering pattern of placental tissue layers. (**D**) The heat map of gene expression in the seven placental layers of Tibetans and Han migrants reveals the same clustering pattern seen in the PCA map. The layers are color-coded (see panel **A**).

The online version of this article includes the following figure supplement(s) for figure 1:

**Figure supplement 1.** The principal component analysis (PCA) plot of the 69 individuals in this study.

**Figure supplement 2.** Comparison of 11 reproductive traits between 35 Tibetans and 34 Han immigrants.

with singleton birth (*Figure 1*). These Tibetan and Han migrant mothers live at the same high altitude (Lhasa, Tibet Autonomous Region, China; elevation = 3650 m), and had experienced their entire pregnancy at this altitude. The population ancestry of the subjects was confirmed by self-claim, and further validated by the genome-sequencing data (Materials and methods, *Figure 1—figure supplement 1*).

We first compared the newborn traits between the Tibetan group and the Han group. As expected, the average BW of the Tibetan newborns is significantly higher than that of the Han newborns (p=0.0121, unpaired Student's t-test), a clear indication of a better fetal development in Tibetans, while there are no significant differences in placental weight (PW) and placental volume (PLV) (*Figure 1—figure supplement 2*). Given the placenta is a highly heterogeneous organs underlying functional and histological specializations in discrete anatomical parts (*Sood et al., 2006*), we dissected each placenta into seven tissue layers according to its anatomic structure (from fetal side to maternal side), including umbilical cord (UC), amnion (AN), chorion (CN), chorionic plate (CP), villus of fetal (VF), villus of intermediate (VI), and villus of maternal (VM), and we conducted RNA-seq of 483 placental samples

(69 placentas × 7 layers) (*Figure 1A*), In total, ~12 billion short reads were generated. After applying a stringent quality control (QC) (Materials and methods), we kept 448 placental transcriptome data, and each sample contains ~27 million reads on average (sample details in *Supplementary file 1a*). In addition, we obtained the genome-wide variants of these mothers and their newborns from our previous studies (*He et al., 2023*; *Zheng et al., 2023*).

As the placenta is a maternal-fetal mosaic organ, we first determined the maternal-fetal compositions of each dissected layer using the genomic and transcriptomic data (see Materials and methods for details). The results show that UC and VF are 100% fetal origin, so are VM, VI, CP, and AN (>95%) with tiny proportions of maternal origin. By contrast, CN is mostly maternal origin (70%) (*Figure 1B*). This pattern is consistent with the anatomic structure of the placenta (*Sood et al., 2006*). Consequently, the RNA-seq data of placenta mostly represent the transcriptomes of the newborns, except for the CN reflecting the mixed transcriptomes of mothers and newborns.

The principal component analysis (PCA) indicates that the placental samples cluster by layers instead of population groups, suggesting that the transcriptomic difference between placental structures is greater than the between-population divergence. This pattern is in line with previous transcriptome of three-layer placentas (CN, AN, and decidua) of Nepali Tibetans, in which the samples were grouped by tissue type instead of ethnicity (*Wu et al., 2022*). Notably, the four trophoblast layers (CP, VF, VI, and VM) cannot be clearly separated in the PCA map and they form a large cluster. These four layers belong to the chorionic villus, the essential structure in placenta (*Wapner and Jackson, 1988*). The UC, AN, and CN layers form three separate clusters (*Figure 1C*). Accordingly, the gene expression heat map analysis generates the same clustering pattern (*Figure 1D*). These results suggest that different layers of the placenta have distinct gene expression profiles, which can provide detailed regulatory information in understanding the role of placenta in fetal development.

## VF and VI show the largest expression divergence between Tibetans and Han

To gain insights into the differences of gene expression in placenta between Tibetans and Han, we performed differential gene expression analysis of the seven placental layers (see Materials and methods for technical details). In total, we identified 579 differentially expressed genes (DEGs) between Tibetans and Han, accounting for 3.4% of the total number of expressed genes. Our results show that VF and VI have the largest numbers of DEGs between Tibetans and Han (305 DEGs in VI and 173 DEGs in VF), accounting for 82.56% (478/579) of the total DEGs in the placenta. By contrast, the numbers of DEGs in the other five layers are much less (8–46 DEGs) (*Figure 2A*, *Supplementary file 1b*). Notably, most of the DEGs (399/579, 68.91%) are detected in only one layer (*Figure 2A*), an indication of a layer-specific differential expression pattern. The results of GO enrichment analyses show that the mRNA regulation metabolic process is up-regulated and immune response (indicated by the terms of defense response to virus) is down regulated in VF of Tibetans compared to Han (*Figure 2B*, *Supplementary file 1c*). In VI, the significant functional terms decreased expression in Tibetans are mostly related to proteins targeting endoplasmic reticulum (ER) (*Figure 2B*, *Supplementary file 1c*). Taken together, these results indicate that the differences between Tibetans and Han are mostly unfolded in the VF and VI layers, and Tibetans have less hypoxia-induced immune response and ER stress in the placenta (*Elvekrog and Walter, 2015*; *Plumb et al., 2015*; *Tenzing et al., 2021*). We did not see significant enrichment terms in the other layers due to the small numbers of DEGs.

There are 85 shared DEGs among the placental layers. As expected, most of them are two-layer-shared DEGs between VF and VI (76/85, 89.41%), and the other 9 are multi-layer-shared DEGs (*Figure 2C*). In particular, there are two four-layer-shared DEGs, and the two involved genes are *KCNE1* (potassium voltage-gated channel subfamily E regulatory subunit 1) and *AC004057.1*. *KCNE1* is significantly up-regulated in all four trophoblast layers (CP, VF, VI, and VM) of Tibetans compared to Han (*Figure 2D*), and it is also the top up-regulated DEG in both VF and VI (*Figure 2E*). It encodes a potassium ion voltage-gated channel protein. The expression of *KCNE1* is reportedly down-regulated in preeclampsia placentas (*Mistry et al., 2011*), and the in vitro experiment also showed its down-regulation in the cultured cells under hypoxia (*Luo et al., 2013*). Therefore, the relative up-regulation of *KCNE1* in Tibetans is presumably beneficial to the reproductive outcomes at high altitude. The function of *AC004057.1* is currently unknown.

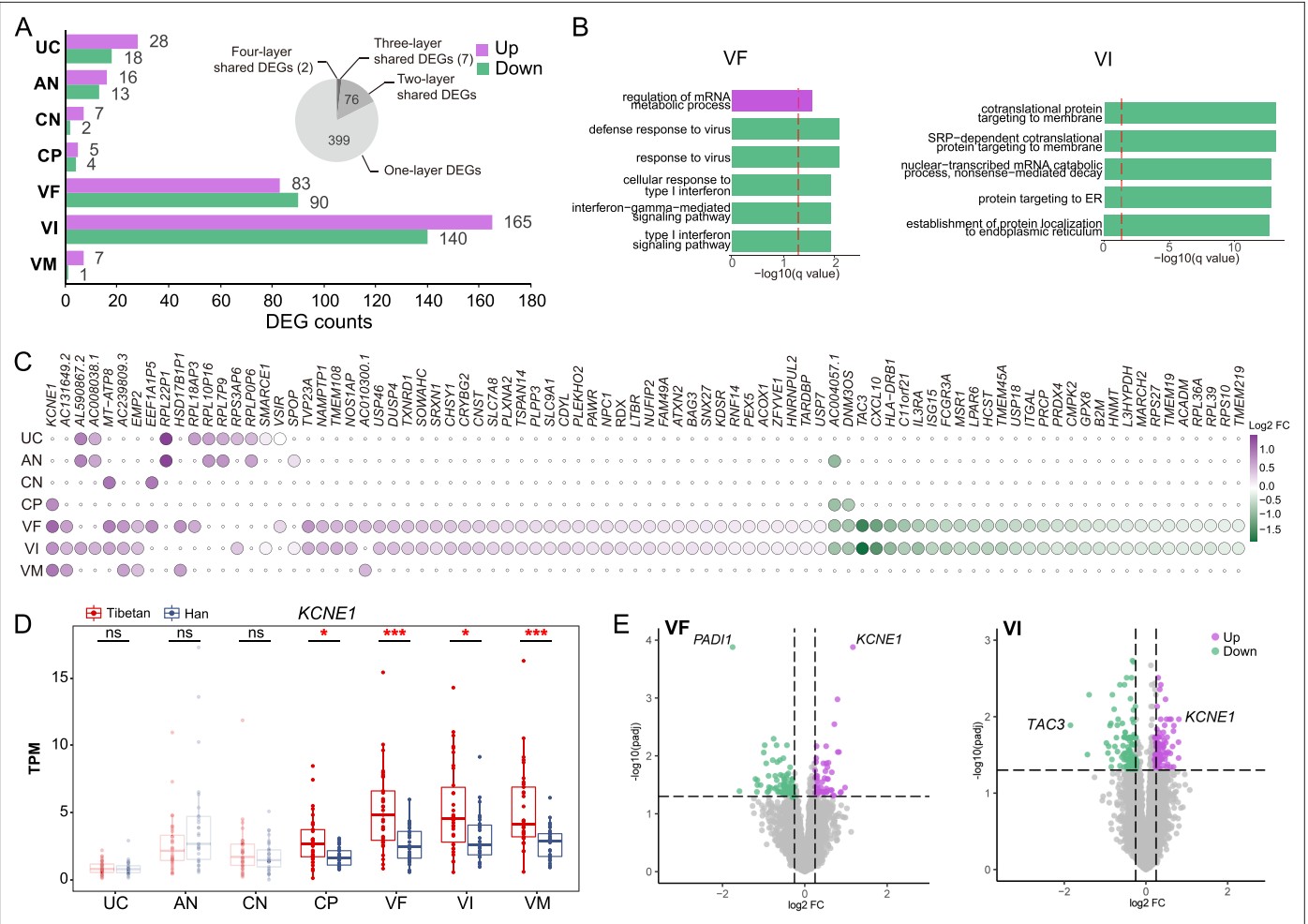

**Figure 2.** The gene expression differences of the placental layers between Tibetans and Han migrants. (**A**) The numbers of differentially expressed genes (DEGs) between Tibetans and Han in each placental layer. Purple: the number of up-regulated DEGs in Tibetans; green: the number of down-regulated DEGs in Tibetans. The pie chart indicates the shared DEGs among two and more placental layers. (**B**) The enriched functional categories (GO terms) of the up-regulated (top) and the down-regulated (bottom) DEGs in villus of fetal (VF) and villus of intermediate (VI), respectively. The dashed line denotes the threshold of significant test (adjusted p-value <0.05). (**C**) The heat map of the 85 shared DEGs among two and more placental layers. (**D**) Comparison of the expression levels of *KCNE1* between Tibetans (n=35) and Han (n=34) in the seven layers of placenta. The significant between-population differences are indicated. The p-value was adjusted by FDR. Adjusted p-value (p): *p<0.05; **p<0.01; ***p<0.001; ns: not significant. For each boxplot, we draw a box from the first quartile to the third quartile. A vertical line goes through the box at the median. The whiskers go from each quartile to the minimum or maximum. (**E**) The volcano plots of the DEGs in VF and VI, respectively. The top genes are indicated.

The online version of this article includes the following figure supplement(s) for figure 2:

**Figure supplement 1.** The top differentially expressed genes (DEGs) of the villus of intermediate (VI) and villus of fetal (VF) layers between Tibetans and Han.

For the down-regulated genes in Tibetans, the top genes in VF and VI are *PADI1* (peptidyl arginine deiminase 1) and *TAC3* (tachykinin precursor 3), respectively (*Figure 2E*). *PADI1* is a VF-specific DEG with 3.4-fold lower expression in Tibetans compared to Han (p=0.0001) (*Figure 2—figure supplement 1*). *PADI1* encodes a member of the peptidyl arginine deiminase family enzymes. The expression of *PADI1* is triggered by hypoxia and it stimulates *PKM2* activity and contributes to the increased glycolysis under hypoxic condition (*Coassolo et al., 2021*). *TAC3* is a VI-VF-shared DEG with both >3-fold changes in the two layers (Tibetans lower than Han, p=0.04 for VI and p=0.04 for VF) (*Figure 2—figure supplement 1*). *TAC3* encodes a member of the tachykinin family of secreted neuropeptides, and its high expression can lead to preeclampsia (*Page et al., 2006*) and severe fetal growth restriction (*Whitehead et al., 2013*). Hence, the observed down-regulation of these two genes suggests that compared to the Han migrants, the placenta of Tibetans is more adapted to

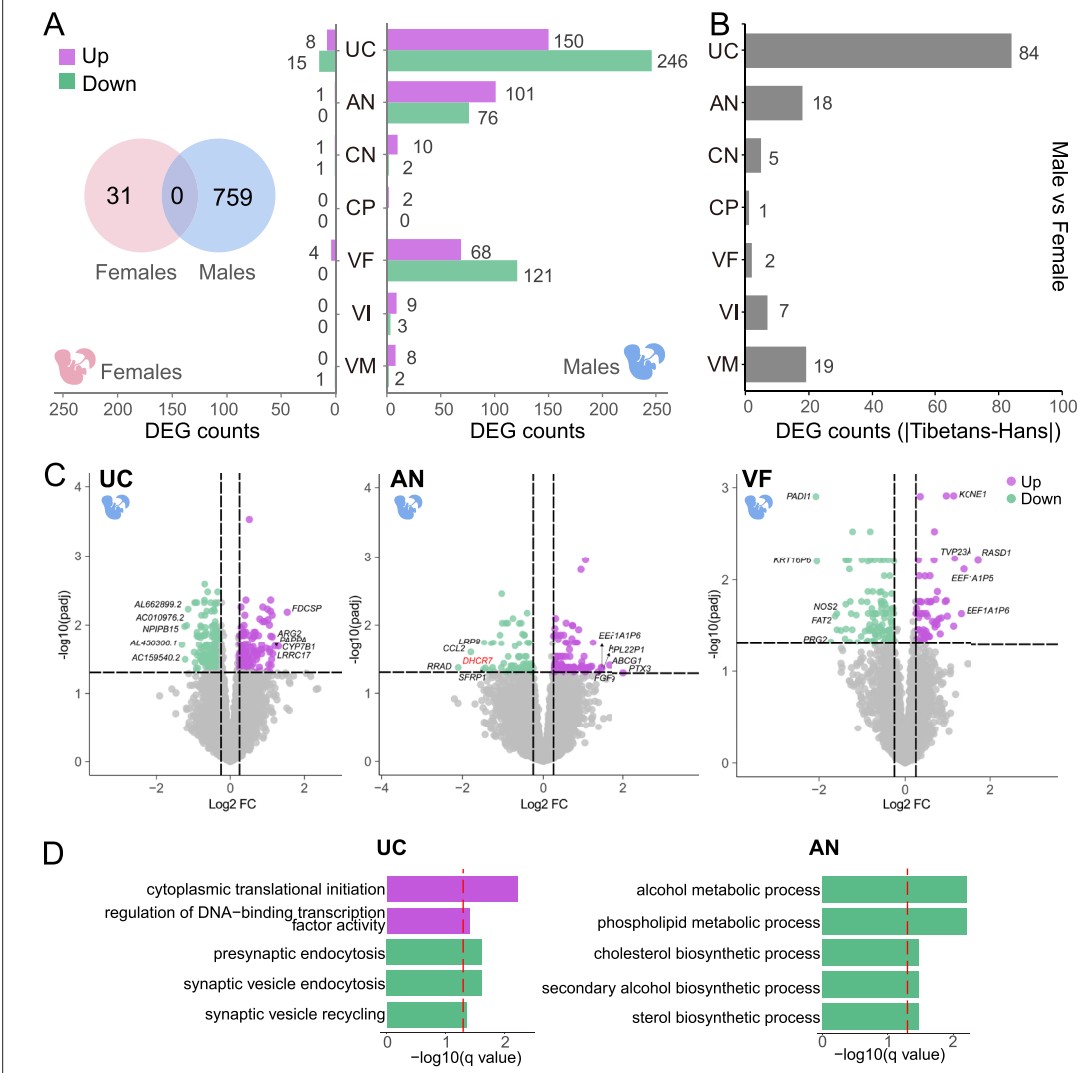

**Figure 3.** Sex-biased placental expression divergence between Tibetans and Han migrants. (**A**) The differentially expressed gene (DEG) numbers of the seven placental layers in the male infants and the female infants, respectively. Left: female (pink); right: male (blue); Venn chart: the overlapped DEGs between male-infant placentas and female-infant placentas. (**B**) Bar plot shows the difference of gene counts between cross-sex DEGs in native Tibetans and Han migrants. (**C**) The up- and down-regulated genes in the male infants of Tibetans compared to Han, shown in the volcano plots of umbilical cord (UC), amnion (AN), and villus of fetal (VF). The top 5 DEGs are indicated. The example gene *DHCR7* is highlighted in red. (**D**) The enriched functional categories (GO terms) of the up-regulated (purple) and the down-regulated (green) DEGs in UC and AN, respectively. The dashed line denotes the threshold of significant test (adjusted p-value<0.05).

The online version of this article includes the following figure supplement(s) for figure 3:

**Figure supplement 1.** The sex-biased gene expression in the placenta.

**Figure supplement 2.** The layer-shared differentially expressed genes (DEGs) in the placentas of males and females.

**Figure supplement 3.** The expression divergence between males and females.

hypoxic environment, presumably with a low level of hypoxia-induced glycolysis and a reduced risk of fetal growth restriction.

## Sex-biased expression divergence in the placenta between Tibetans and Han

Given the male fetuses are more sensitive to high-altitude hypoxia than the female fetuses as described in our previous report (*He et al., 2022*), we next analyzed the placental transcriptomes by taking into account the infant gender (*Figure 3*, Materials and methods). When the gender identity

is superimposed onto the PCA map in *Figure 1D*, we did not see any clustering patterns by genders, implying that the gender-related expression changes do not cause a dramatic profile shift (*Figure 3— figure supplement 1*). However, surprisingly, we found a striking difference in view of the numbers of DEGs (Tibetans vs. Han), where the male-fetus placentas have many more DEGs than the female-fetus placentas (759 genes vs. 31 genes), and there is no intersection between them (*Figure 3A*). This between-sex difference is also reflected by the shared DEGs between the gender-combined DEG set and the gender-separated DEG sets, where the male set has 128 shared DEGs, but only 5 shared DEGs in the female set (*Figure 3—figure supplement 1*), suggesting that the male-fetus placentas contribute more to the detected between-population expression divergence.

Of the 759 male-fetus DEGs, 728 DEGs are detected in one layer. There are 25 two-layer-shared DEGs, 4 three-layer-shared DEGs, and 2 four-layer-shared DEGs (*Figure 3—figure supplement 2*). By contrast, there is no multi-layer-shared DEGs in the female-fetus placentas. Among the 31 layer-shared DEGs in the male-fetus placentas, 22 DEGs are up-regulated and 9 DEGs are down-regulated in Tibetans. Similar to the above results, there are a large portion of overlap between the layer-shared DGEs of the male-fetus placentas and the gender-combined placentas (*Figure 3—figure supplement 2*).

Layer-wise, in the male-fetus placentas, UC, AN, and VF have the largest numbers of DEGs (396, 177, and 189, respectively, *Figure 3A*, *Supplementary file 1d*), accounting for 95.49% (762/798) of all DEGs, while the DEGs in the other layers are much less (<15). By contrast, the great majority of the DEGs in the female-fetus placentas is from UC (23 genes, *Supplementary file 1e*), while the other layers only have a few DEGs (<5). This DEG pattern is highly different from that seen in the gender-combined result (*Figure 2A*), again, an indication of a sex-biased expression divergence between Tibetans and Han migrants at high altitude. Similar with the gender-combined result, the great majority (>95% in both males and females) of the DEGs are one-layer DEGs. Given the previous studies have been mostly focused on the functional role of the placental villi (*Lorca et al., 2021*; *Tana et al., 2021*; *Vaughan et al., 2020*), our gender-separated analyses illustrate the importance of UC and AN in fetal development, especially for the male fetus.

Markedly, in the gender-combined result, we only detected 31 DEGs in the UC layer, contrasting the large number of DEGs (396 genes) in the male-fetus placentas. Considering 62.63% of DEGs (248/396) with an opposite direction of between-population expression divergence in males and females, respectively (*Figure 3—figure supplement 3*), we reckon that there might be other factors such as sample size or cell composition affecting the identification of DEGs, which could cancel out the differences in the gender-combined analysis. Consistently, the between-sex expression comparison (DEGs between genders in a population: Tibetan male vs. female; Han male vs. female) indicates that the UC shows the largest DEG count difference between native Tibetans and Han migrants (*Figure 3B* and *Figure 3—figure supplement 3*). Consequently, the UC layer shows the most pronounced gender-dependent expression divergence.

The sex-biased expression divergence in placenta is likely caused by the differential responses of male and female fetuses to hypoxic stress at altitude, as proposed in our previous study (*He et al., 2022*). In UC, AN, and VF, where the largest numbers of DEGs were detected in the male placentas, the top 5 DEGs are indicated in *Figure 3C*. These genes are primarily involved in immune responses, embryonic development, cell migration, and cholesterol metabolism, all of which are closely related to fetal development (*Barlan et al., 2017*; *Choi et al., 2019*; *Sifakis et al., 2018*; *Yang et al., 2021*). In UC, the enriched GO terms include up-regulation of cytoplasmic translational initiation and DNA-binding transcription factor activity, and down-regulation of presynaptic endocytosis in Tibetans (*Figure 3D*, *Supplementary file 1c*). In AN, Tibetans show a down-regulation in alcohol metabolic-related process, phospholipid metabolic process, and cholesterol biosynthetic process (*Figure 3D*, *Supplementary file 1c*). These enriched functional categories reflect a more active protein synthesis and a reduced risk of hypoxia-induced metabolic disturbance, which are presumably beneficial to fetal development of the Tibetan male newborns (*Guzel et al., 2017*; *Yung et al., 2008*; *Zhang et al., 2017*). We did not see significant functional terms in VF (*Supplementary file 1c*).

It is known that hypoxia can lead to inhibition of global protein synthesis and activation of ER stress (*Koumenis, 2006*). Our results suggest that compared to native Tibetans, the Han migrants are more responsive to hypoxic stress, reflected by their relative down-regulation of translational initiation and transcription factor activity, and this trend is mostly exemplified in the UC layer of the male placentas

since UC is the key channel of maternal-fetal exchange of oxygen and nutrition. On the other hand, cholesterol plays an important role in embryonic development (*Cortes et al., 2014*). We detected the down-regulation of a cholesterol regulating gene (*DHCR7*) in Tibetans (*Figure 3C*). The *DHCR7*-encoded protein can covalently link to *Shh* (sonic hedgehog) to participate in brain, limb, and genital development (*Roux et al., 2000*). Accumulation of *DHCR7* and deficiency in cholesterol production will cause a devastating developmental disorder (*Prabhu et al., 2016*).

## Gene expression changes in the UC are associated with BW of the male newborns

As a critical organ for maternal-fetal nutrition and oxygen exchanges, the physiological status of the placenta is closely related to the neonatal status. We performed weighted gene co-expression network analysis (WGCNA) to investigate the association between gene expression modules and the newborn-related traits. The investigated traits include BW, biparietal diameter (BPD), femur length

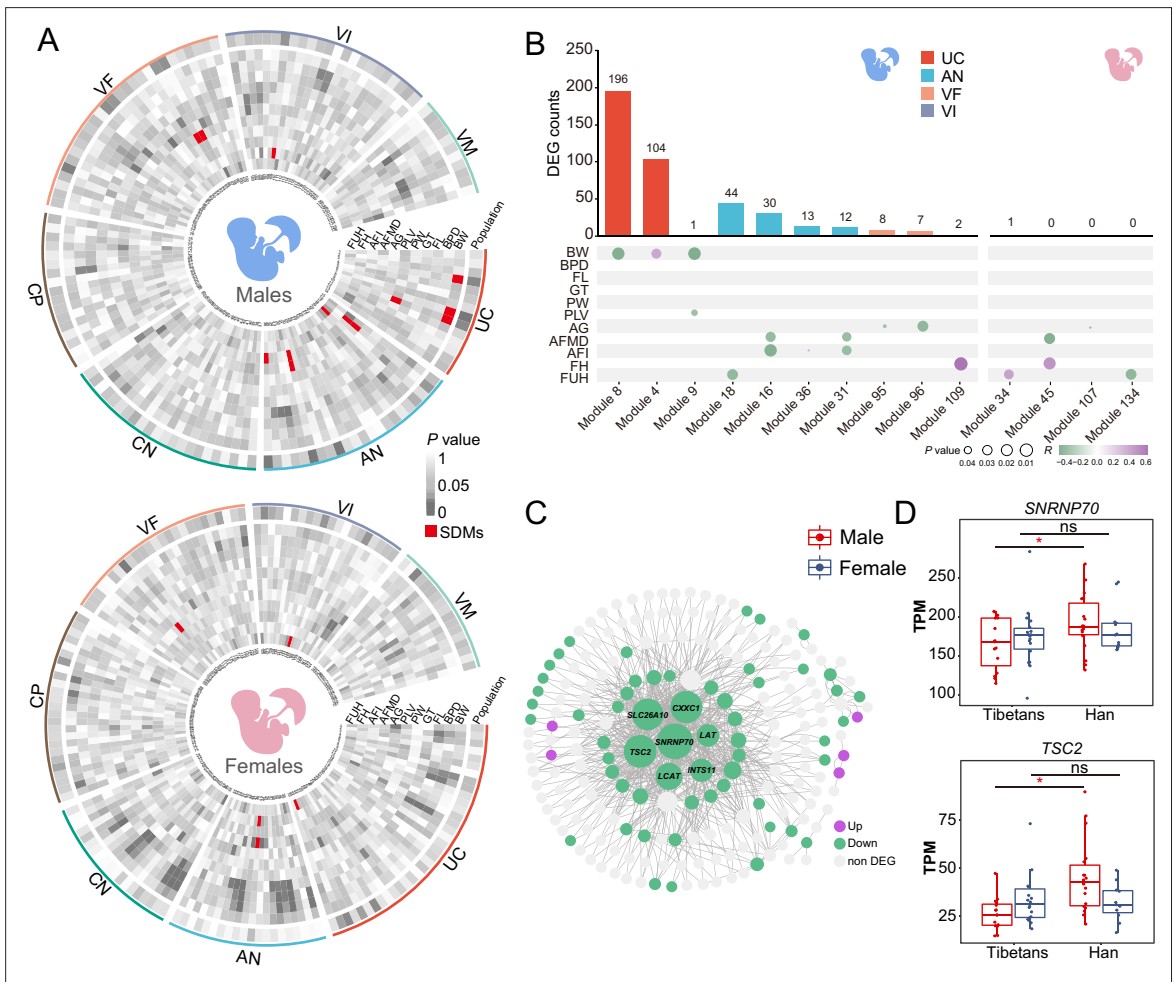

**Figure 4.** Gene expression modules in umbilical cord (UC) of the male infants correlate with neonatal phenotypes. (**A**) The heat maps of p-values showing the correlations between gene expression and the newborn traits of the male-infant placentas (left) and the female-infant placentas (right). Inner ring: module name; outer ring: layers; red: significantly differential modules (SDMs). (**B**) The counts of the module-associated differentially expressed genes (DEGs) in the seven placental layers of the male infants and the female infants (the upper panels), and their correlations with the newborn traits (the lower panels). (**C**) The gene co-expression network of Module 8 of the male infants. (**D**) Gene expression comparisons of two hub genes in UC of the male infants between Tibetans (n: male=16,female=19) and Han (n: male=21,female=13). The p-value was adjusted by FDR. Adjusted p-value (p): *p<0.05; ns: not significant. For each boxplot, we draw a box from the first quartile to the third quartile. A vertical line goes through the box at the median. The whiskers go from each quartile to the minimum or maximum.

The online version of this article includes the following figure supplement(s) for figure 4:

**Figure supplement 1.** The sex-biased correlation between gene expression and traits of the male-infant and female-infant placentas.

(FL), gestation time (GT), PW, PLV, abdominal girth (AG), amniotic fluid maximum depth (AFMD), amniotic fluid index (AFI), fetal heart (FH), and fundal height (FUH).

A total of 136 and 161 molecular modules were obtained from the placentas of the male infants and the female infants, respectively. We define a module as a significantly differential module (SDM) when the gene expression of a module is significantly correlated to population divergence (Tibetans vs. Han, p<0.05) and the newborn-related traits (p<0.05) (see Materials and methods for details). Totally, there are 10 SDMs in males, but only 4 in females (*Figure 4A*, *Supplementary file 1f, g, h, i*, Materials and methods). Markedly, the UC layer of the male infants has 3 SDMs, including Module 4 (R=0.39, p=0.03), Module 8 (R=–0.45, p=8.6 × 10$^{-3}$), and Module 9 (R=–0.47, p=6.4 × 10$^{-3}$) (*Figure 4A*), which are significantly associated with two newborn traits (BW and PLV), supporting a close involvement of UC in fetal development, especially for the male fetuses. There are four AN SDMs, two VF SDMs, and one VI SDM that are associated with five newborn traits, including AFI (Module 16, Module 31, and Module 36), AFMD (Module 16 and Module 31), FUH (Module 18), AG (Module 95 and Module 96), and FH (Module 109), respectively (*Figure 4A*). For the female placentas, four SDMs are associated with AG (Module 107), AFMD (Module 45), FH (Module 45), and FUH (Module 34), respectively (*Figure 4A*). These identified SDMs suggest that the difference of transcriptional regulation of the placental layers between Tibetans and Han migrants do have impact on the newborn status, especially on the male infants.

We next looked at the DEGs in the SDMs. For the male placentas, there are a lot of DEGs involved in the 10 SDMs. By contrast, among the four SDMs in the female placentas, there is only one DEG (*Figure 4B*). This pattern is consistent with the male-biased gene expression divergence between Tibetans and Han. In particular, in the UC layer of the male placentas, there are 196 DEGs in Module 8 and 104 DEGs in Module 4. Most of the DEGs in Module 8 are down-regulated in the UC layer of Tibetans (*Figure 4C*), and this module is negatively correlated with BW (*Figure 4B* and *Figure 4— figure supplement 1*), indicating that the down-regulated gene expression module in the UC layer of the Tibetan male placentas may contribute to the higher BW of the Tibetan newborns compared to Han. There are seven hub genes in Module 8, and all of them are down-regulated in the Tibetan male placentas (*Figure 4C*). For example, *SNRNP70* (small nuclear ribonucleoprotein U1 subunit 70) enables U1 snRNA-binding activity, and it is one of the known hypoxia-inducible target genes (*Kurihara et al., 2016*). *TSC2* (TSC complex subunit 2) is a tumor suppressor gene that encodes the growth inhibitory protein tuberin. Inhibition of the mTOR pathway by hypoxia requires *TSC2*, leading to cell proliferation under hypoxia (*Brugarolas et al., 2004*; *Figure 4D*). Hence, the down-regulation of the great majority of DEGs in Module 8 suggests that the male fetuses are likely more sensitive to hypoxic stress than the female fetuses, and compared to native Tibetans, they are more responsive in the Han migrants, which is associated with the higher BW in the adapted natives (Tibetans) compared to the acclimatized migrants (Han). Taken together, the above results indicate that the UC layer plays a key role in the development of the male fetus and ultimately affects BW.

*GHR* (growth hormone receptor) is the only DEG in the SDMs of the female newborns (*Figure 4B*). The gene expression level of *GHR* in the VF layer is positively correlated with FUH (R=0.44, p=0.017) of the female newborns, an indicator of fetal growth (*Morse et al., 2009*), but not the male newborns (*Figure 4—figure supplement 1*). Consistently, it is up-regulated in Tibetans, corresponding to the higher FUH in Tibetans (*He et al., 2023*). Interestingly, the expression pattern of *GHR* mirrors *GH* (growth hormone) in placenta early development and embryonic growth (*Harvey and Baudet, 2014*).

## Histological outcomes of the sex-biased expression divergence

To further test the impact of the observed sex-biased expression divergence, we performed histological examination of the UC, VF, and VI where the most prominent sex-biased expression and the largest numbers of DEGs were detected (*Figure 5A*). We analyzed 20 individual placenta samples (male-infants: 5 Tibetans vs. 5 Han; female-infants: 5 Tibetans vs. 5 Han). For the UC, five parameters of UC were evaluated using cross-sections, including umbilical vein lumen (UVL), umbilical vein wall (UVW), umbilical artery lumen (UAL), umbilical artery wall (UAW), and umbilical artery intima and media (UAIM) (Materials and methods).

In line with the expression data, we observed an obvious sex-biased histological differences in UC, and the values of UVW and UAIM are significantly larger in Tibetan male infants than Han (p=0.003 and p=0.03), but no differences seen in the female infants (p>0.05) (*Figure 5B and C*). The

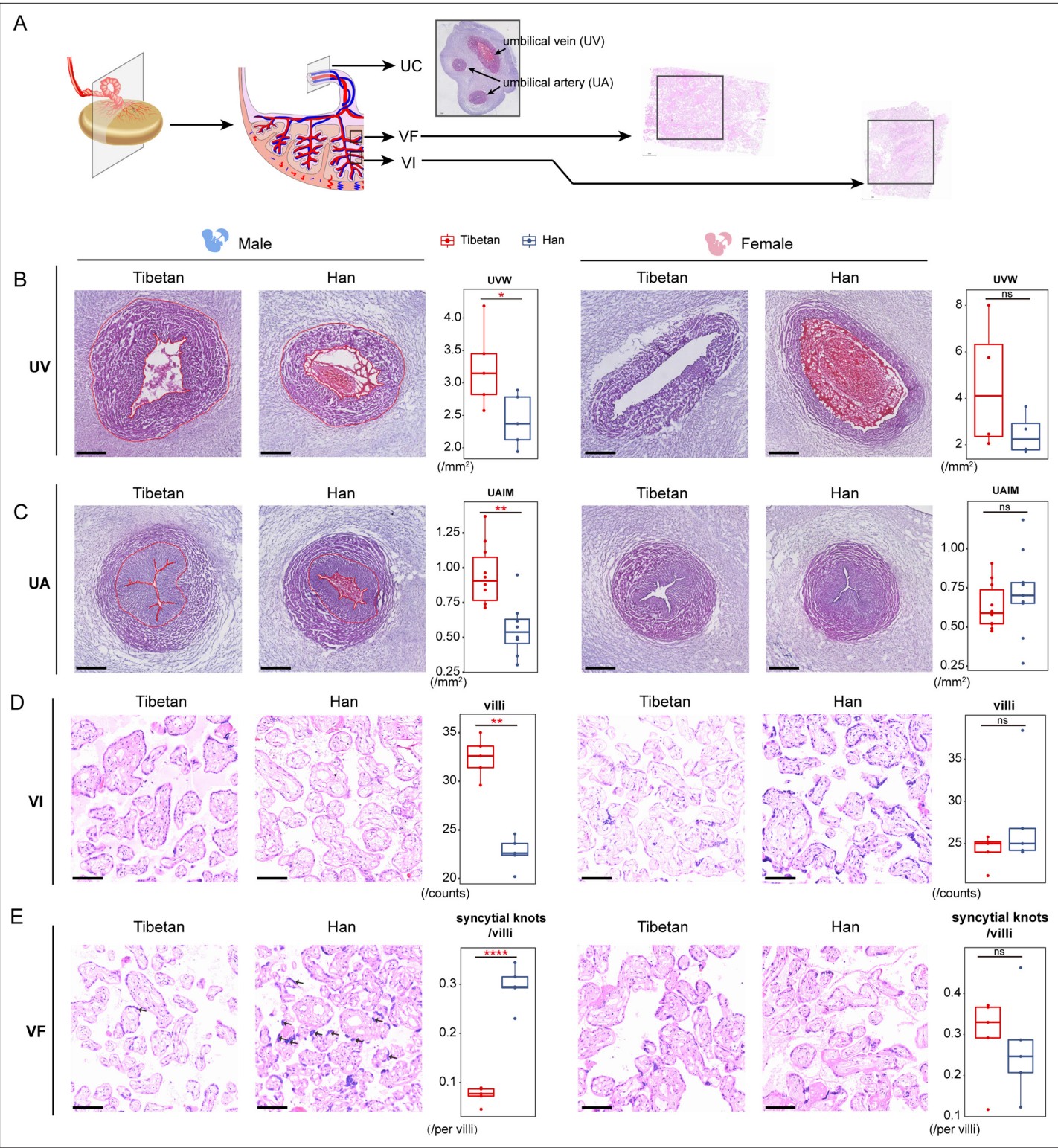

**Figure 5.** Sex-biased histological changes in the male-infant placentas of Tibetans. (**A**) Schematic diagram of the sampling strategy in histological analysis. (**B, C**) The H&E stained cross-sections of the umbilical cord (UC) vessels (umbilical vein [UV], panel **B**; and umbilical artery [UA], panel **C**). The umbilical vein wall (UVW) and umbilical artery intima and media (UAIM) are highlighted in red. The boxplots show the comparisons of area size of UVW and UAIM between Tibetan (n=5) and Han (n=5) male infants (left) or female infants (right). The two umbilical arteries of each UC were subject to analysis in panel C. Scale bar: 500 μm. (**D, E**) The H&E stained sections of the placental villus of intermediate (VI) and villus of fetal (VF). The arrows denote the syncytial knots. The boxplots show the comparison of the villi numbers or the syncytial knot/villi ratios between Tibetan (n=5) and Han

*Figure 5 continued on next page*

Figure 5 continued

(n=5) male infants (left) or female infants (right). Unpaired Student's t-tests are used to evaluate the significance of difference. Scale bar: 100 μm. *p-value<0.05; **p-value<0.01; ns: not significant. For each boxplot, we draw a box from the first quartile to the third quartile. A vertical line goes through the box at the median. The whiskers go from each quartile to the minimum or maximum.

The online version of this article includes the following figure supplement(s) for figure 5:

**Figure supplement 1.** Histological outcome of the sex-biased structure divergence in the placenta.

previous study reported a significant association between preeclampsia and a smaller thickness and wall area of the umbilical vein and artery, independent of GT and BW (*Herzog et al., 2017*), which explains the presumably adaptive histological changes of UC (increased UVM and UAIM) in Tibetans. No significant differences between Tibetans and Han were observed in the other UC parameters (*Figure 5—figure supplement 1*).

For placental trophoblast (VF and VI), we counted the number of villi and syncytial knots in the placenta (Materials and methods). In the male placentas, Tibetans have more villi in the VI layer than Han (p<0.0001, *Figure 5D*), and the same trend was seen in the VF layer though not significant (p=0.13, *Figure 5—figure supplement 1*). Moreover, when we counted the syncytial knots in the villi, we saw a significantly lower ratio of syncytial knots/villi in male placenta of Tibetans (p<0.0001, *Figure 5E*), an indication of healthier villi because the appearance of syncytial knots are the degenerating feature of placenta with remarkable accumulations of nuclei and degenerating organelles (*Heazell et al., 2007*). By contrast, no difference was detected in the placentas of the female newborns in view of the number of villi and the ratio of syncytial knots/villi in either the VI layer or the VF layer (*Figure 5D and E* and *Figure 5—figure supplement 1*).

These sex-biased histological results are consistent with the patterns seen in gene expression. The male infants show remarkably more DEGs between native Tibetans and Han migrants than female infants, especially those genes involved in immune responses. Also, our results prove that such male-biased expression divergence is closely associated with BW. Therefore, as a distinctive remodeling of the UC in male infants, the male-biased histological differences between Tibetans and Han Chinese are most likely induced by the different inflammatory responses and intrauterine hemodynamics. For placenta trophoblast, the difference of the syncytial knots/villi ratios between VF and VI may be caused by their differences in the distribution of placental villi. Branches of umbilical artery pass from the CP layer before producing stem villi inferiorly (*Castellucci et al., 1990*). The VF layer is next to the CP layer and the increased villus number in Tibetans suggests that the stem villi extending out from umbilical artery produce more villous branches, which is presumably beneficial to the maternal-fetal material exchange in Tibetans. Consistently, the reduced number of syncytial knots in the VI layer of Tibetans might be conducive to a better material exchange efficiency (*Fogarty et al., 2013*).

## Natural selection acts on the placental DEGs with expression divergence between Tibetans and Han

Given the overall gene expression divergence in the placenta between Tibetans and Han migrants can explain the better newborn traits of Tibetans, the involved genes could be the target of natural selection. At the same time, most of the adaptive variants identified in Tibetan population are located in the noncoding regions of the genome, suggesting the functional outcomes of these genetic variations are likely achieved by gene expression regulation.

To see whether the detected DEGs of placenta tissues are subject to Darwinian positive selection, therefore contributing to the genetic adaptation of reproductive fitness in Tibetans, we checked all the identified DEGs and see whether they were listed in the 192 reported Tibetan selection-nominated genes (TSNGs), i.e. the genes showing signatures of Darwinian positive selection in Tibetans (*Zheng et al., 2023*). In total, we found that 13 DEGs are TSNGs, including 4 DEGs from the gender-combined gene list and 9 DEGs from the male-only gene list. No DEGs from the female-only gene list is overlapped with the TSNGs (*Figure 6A*). Permutation test shows that the overlapped male DEGs with TSNGs are over-represented (p<1e-4) (*Figure 6—figure supplement 1*, Materials and methods). Among the four DEGs with selection signals in the gender-combined gene list, we saw *EPAS1*, the gene with the strongest signal of selection in Tibetans, and it shows a significant down-regulation in the UC layer of Tibetans compared to Han migrants (p=0.04) (*Figure 6B*). Previously, there were

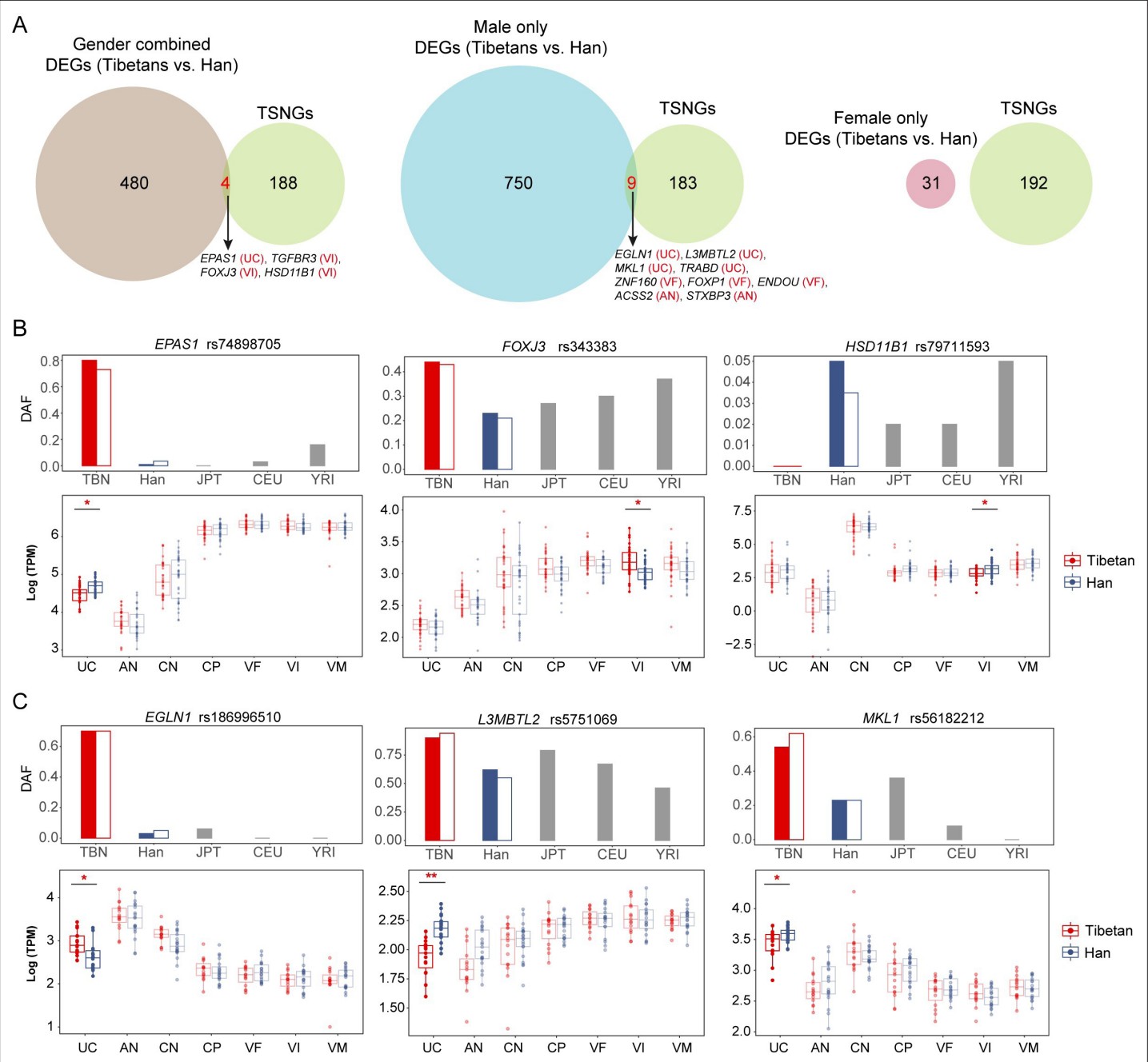

**Figure 6.** The differentially expressed genes (DEGs) in the placenta that underwent positive selection in Tibetans. (**A**) The Venn plots show the intersections between the placental DEGs and the 192 Tibetan selection-nominated genes (TSNGs), covering the gender-combined and the gender-separated DEGs. (**B**) The three DEGs (gender-combined) under positive selection in Tibetans. The upper panels show the allele frequencies of variants with the strongest signals of selection within the gene. Besides Tibetans and Han Chinese, the data from three other reference populations are also presented, including Japanese (JPT), Europeans (CEU), and Africans (YRI) from the 1000 Genome Project. The solid and hollow bars in red denote the allele frequencies in the published 1001 Tibetan individuals and 35 Tibetan individuals, respectively, and the solid and hollow bars in blue denote the allele frequencies in the published 103 Han individuals and 34 Han individuals, respectively. The bottom panels show the comparison of the expression levels between Tibetans (n=35) and Han (n=34) in the seven layers of placenta. Only the significant between-population differences are indicated. The p-value was adjusted by FDR. Adjusted p-value: *p-value<0.05; **p-value<0.01; ***p-value<0.001. For each boxplot, we draw a box from the first quartile to the third quartile. A vertical line goes through the box at the median. The whiskers go from each quartile to the minimum or maximum. (**C**) Expression levels of three DEGs (sex-separate analysis) with positive selection signals in seven placental layers of Tibetan males (n=16) and Han males (n=21).

The online version of this article includes the following figure supplement(s) for figure 6:

*Figure 6 continued on next page*

*Figure 6 continued*

**Figure supplement 1.** The distribution of the overlapped genes between the identified differentially expressed genes (DEGs) and the 192 randomly selected genes based on 10,000 permutations.

**Figure supplement 2.** The correlation between the expression of FOXJ3 and birth weight (BW)/placental weight (PW) in the chorion (CN) layer of placenta.

**Figure supplement 3.** The placental differentially expressed genes (DEGs) of the amnion (AN) and villus of fetal (VF) layers with signals of positive selection in Tibetan population.

---

multiple SNPs of *EPAS1* with reported GWAS signals for BW in lowland populations with a large sample size (N>300,000) (*Warrington et al., 2019*). Therefore, the down-regulation of *EPAS1* in the Tibetan placentas likely reflects a blunted hypoxic response that may improve vasodilation of UC for better blood flow, and eventually leading to the higher BW in Tibetans.

*FOXJ3* and *HSD11B1* are the other two genes overlapped between the placenta DEGs and the TSNG set (*Figure 6B*). *FOXJ3* (forkhead box J3) is a transcriptional activator, and reportedly plays an important role in spermatogenesis (*Ni et al., 2016*). We observed a significant correlation of *FOXJ3* expression of the CN layer with placenta weight and BW (p=0.02 and p=0.004, respectively) (*Figure 6—figure supplement 2*), suggesting a potential role of *FOXJ3* in transcription regulation underlying early fetal development. *HSD11B1* (hydroxysteroid 11-beta dehydrogenase 1) is a micro-somal enzyme that catalyzes the conversion of the inactive hormone cortisol to the active metabolite cortisone. The Tibetan placentas show a lower *HSD11B1* expression than Han migrants (p=0.01), which may bring a lower cortisol stress hormone level. Given the hypoxia-induced glucocorticoid elevation can have impact on placental vascular development and nutrient transport (*Ozmen et al., 2017*), the lower cortisol stress hormone level in the Tibetan placentas might be advantageous under hypobaric hypoxia at high altitude.

Among the nine DEGs with selection signals from the male-only gene list, four are DEGs of the UC layer, including *EGLN1*, *L3MBTL2*, *MKL1*, and *TRABD* (*Figure 6C*, *Figure 6—figure supplement 1*, and *Supplementary file 1j*). *EGLN1* is one of the TSNGs with strong signals of selection in Tibetans, and it is a member of the HIF (hypoxia-induced factor) pathway and works together with *EPAS1* (*Peng et al., 2011*). There are two Tibetan-enriched missense mutations in the *EGLN1* gene, and the mutations are reportedly related to the lower erythropoiesis and augmented hypoxic ventilatory response in Tibetans (*Lorenzo et al., 2014*; *Song et al., 2020*). *EGLN1* is temporally expressed in an oxygen-dependent fashion, with the greatest mRNA expression at 10–12 weeks of gestation (*Ietta et al., 2006*). In the UC layer, we detected a significant down-regulation of *EGLN1* in the Tibetan male placentas (p=0.01), but not in the Tibetan female placentas (p=0.91) (*Figure 6C*). Among the other three genes, *MKL1* (myocardin-related transcription factor A, also named *MRTFA*) was reportedly involved in vasoconstriction mediated by human vein endothelial cells under hypoxia (*Yang et al., 2013*), and it possesses a series Tibetan-enriched SNPs and structural variants (*He et al., 2020*; *Zheng et al., 2023*). The rat gene knockout model indicated that an *MKL1* deletion could improve hypoxia-induced pulmonary hypertension (*Yuan et al., 2014*), and the Tibetan-enriched 163 bp deletion at *MKL1* is associated with the lower pulmonary arterial pressure of Tibetans (*He et al., 2020*). There-fore, the down-regulation of *MKL1* in the male placentas likely works in the same way as *EGLN1* and *EPAS1* do, and may help reduce the risk of hypoxia-induced hypertension in Tibetans. In addition, *L3MBTL2* (L3MBTL histone methyl-lysine binding protein 2) is involved in DNA damage response (*Nowsheen et al., 2018*), and cell proliferation and differentiation (*Markus et al., 2003*). The function of *TRABD* (TraB domain containing) is not known. More functional data are needed to understand the adaptive roles of these genes in fetal development at high altitude. Except for the DEGs of the UC layer, two DEGs of the AN layer (*ACSS2* and *STXBP3*) and three DEGs of the VF layer (*ZNF160*, *FOXP1*, and *ENDOU*) are also overlapped with the TSNG set (*Figure 6—figure supplement 3*). These genes are mainly involved in the functions about embryonic development, cardiac valve morphology, and immune response (*Hu et al., 2006*; *Kanda et al., 2005*).

## Discussion

The placenta is the key organ for fetal development, and the transcriptional profiles of the placenta are highly informative in revealing the gene expression patterns and the regulatory networks, which

eventually contribute to the newborn health. In this study, with a placental transcriptomic and histological profiling of native Tibetans and Han migrants living at the same high altitude, we intend to dissect the molecular mechanism of the observed higher reproductive fitness of Tibetans and to understand how the gene expression regulation in the placenta explains the genetic adaptation of Tibetans in view of fetal development, the determinant outcome of natural selection.

We show that among the seven anatomic layers of the placenta, VF and VI display the largest expression divergence between Tibetans and Han migrants, which is expected given the known crucial role of the placental trophoblast in fetal development (*Kingdom et al., 2000*). The down-regulated DEGs in VF and VI of Tibetans (compared to Han) are mainly involved in immune response and ER stress. It is known that besides the maternal-fetal exchanges of nutrient, oxygen, and waste, the placenta is also an important immune organ, and the immune response is one of the physiological reactions under chronic hypoxia (*Facco et al., 2005*). Placental hypoxia/ischemia can lead to the release of a variety of placental factors, the activation of circulating immune cells, and autoantibodies, and have a profound impact on blood flow and arterial pressure regulation (*Zárate et al., 2014*). Hence, our results suggest that Tibetans might acquire the ability of overcoming the hypoxia-induced inflammation and ER stress protein translation inhibition at high altitude.

It is known that the sex of the fetus has a detectable impact on the developmental status (*Clifton, 2010*), and maternal hypoxia could alter placental formation in a sex-specific manner in rodents (*Cuffe et al., 2014*). In addition, sex differences in placental traits are reportedly associated with high elevation adaptation in Andeans (*Jackson et al., 1987*). However, previous studies on human placenta did not take the fetus's gender into account in gene expression analyses. In this study, by looking at the transcriptomic profiles of the male and female placentas separately, we discovered a striking pattern of a male-biased expression divergence between Tibetans and Han migrants. By contrast, there is almost no difference in placental gene expression of the female newborns between Tibetans and Han migrants. This result suggests that the male placenta is more sensitive to environmental stresses such as hypobaric hypoxia at altitude, and consequently, the expression divergence between Tibetans and Han migrants is more pronounced in males.

In particular, the UC layer possesses the largest number of DEGs (396 genes) in the male placentas, and the enriched functional categories of these DEGs imply a more active protein synthesis and a reduced risk of hypoxia-induced metabolic disorder in Tibetans. UC is the main channel of maternal-fetal exchange. Previous studies have shown that UC is important for blood vitamin D (*Alp et al., 2016*) and oxygen transportation (*Postigo et al., 2009*; *Yancey et al., 1992*). We speculate that UC may play a key role in the differential development of the male fetus between Tibetans and Han migrants, reflected by the three detected male-specific molecular modules that ultimately affect BW. In addition, in the UC layer, the between-sex expression comparison indicates that the expression differences between the male placentas and the female placentas are more pronounced in Han migrants (117 DEGs) compared to Tibetans (33 DEGs) (*Figure 3—figure supplement 3*). Hence, under hypoxic stress, besides the known important functions of the trophoblast layers of the placenta, UC and the other non-trophoblast layers are also crucial, especially for the development of the male fetus. This observation serves as an important guiding information for future placental research.

Furthermore, at the histological level, we also see the male-biased differences between Tibetans and Han migrants (*Figure 5*). The UC and VI/VF show distinctive histological changes only in the male infants, the phenotypic outcome of the observed sex-biased expression divergence between Tibetans and Han migrants. The UC and fetal vasculature share the same embryonic origin, and vessels are often used as an important index reflecting newborn vascular health (*Jin and Patterson, 2009*). We observed that the UAW and artery intima and media are significantly larger in the Tibetan male infants than in Han, consistent with the previous report that native Tibetans can prevent hypoxia-associated IUGR accompanied with the higher uteroplacental blood flow in pregnancy (*Moore et al., 2004*). It is known that the UC is highly influenced by systemic and local hemodynamic conditions of pregnancy, such as blood flow, oxygen tension, and oxidative stress (*Blanco et al., 2011*). Preeclampsia is a complex disease characterized by an increased maternal blood pressure during pregnancy and increased risks of fetal growth restriction and preterm birth (*Duley, 2009*). It was reported that environmental hypoxia of high altitude impairs fetal growth, increases the incidence of preeclampsia, and, as a result, significantly increases the risk of perinatal and/or maternal morbidity and mortality (*Julian, 2011*). Additionally, the histological and morphological studies reported that the preeclampsia is

associated with a smaller UC vein area and wall thickness (*Herzog et al., 2017*; *Inan et al., 2002*). Therefore, our observation of the larger UAW and UAIM suggests that native Tibetans might benefit from a remolding of the UC vessels, due to the decreased expression of inflammation-related genes in the Tibetan placentas. Eventually, Tibetans have achieved a higher uteroplacental blood flow, a lower prevalence of preeclampsia, a higher newborn BW, and a lower mortality at high altitude. Notably, all these adjustments are more significant in the male infants than in the female infants.

The adaptive changes are also reflected in the placental trophoblast. The VF and VI layers are the substantial parts of placental villi, which are composed of trophoblastic villi and are responsible for the exchange of nutrients and active transportation. Syncytial knot is a specialized structure of trophoblastic villi, which increases with fetal age and is a feature of placenta maturity (*Loukeris et al., 2010*). It is reportedly increased in the placenta of eclampsia and fetal growth restriction. In vitro experiments have proved that syncytial knots can be induced by hypoxia and other stressful conditions (*Heazell et al., 2007*). In the Tibetan male placentas, there are fewer syncytial knots compared to Han migrants, likely resulting from the observed male-biased expression changes. More functional data is needed to establish the mechanistic connection between the expression profiles and the histological outcomes.

As the gene with the strongest signal of natural selection in Tibetans, *EPAS1* has been reported in numerous studies on its contribution to high altitude adaptation. In this study, we detected a significant expression reduction of *EPAS1* in the Tibetan UC compared to the high-altitude Han. It was reported that the selected-for *EPAS1* variants/haplotype were associated with lower hemoglobin levels in the Tibetan highlanders with a major effect (*Beall et al., 2010*; *Peng et al., 2017*), and the low hemoglobin concentration of Tibetans is causally associated with a better reproductive success (*Cho et al., 2017*; ). Therefore, we speculate that the selective pressure on *EPAS1* is likely through its effect on hemoglobin, rather than directly on the reproductive traits. The down-regulation of *EPAS1* in placenta likely reflects a blunted hypoxic response that may improve vasodilation of UC for better blood flow, and eventually leading to the higher BW in Tibetans (*He et al., 2023*). For *EGLN1*, another well-known gene in Tibetans, we detected between-population expression difference in the male UC layer, but not in other placental layers. Considering the known adaptation mechanism of *EGLN1* is attributed to the two Tibetan-enriched missense mutations, the contribution of *EGLN1* to the gene expression change in the Tibetan UC was unexpected and worth to be explored in the future.

It should be noted that there are limitations in our study. We only analyzed the full-term placentas. At this stage, the placenta has fully matured and we may not be able to capture the dynamic changes during earlier placental and fetal development. Also, we did include placental data of a low-elevation reference, and we could not completely rule out the possible effect of random genetic drift on gene expression divergence between populations, which is not due to adaptation in Tibetans. Additionally, although we have identified four DEGs (Tibetans vs. Han migrants) with reported signals of Darwinian positive selection (*Zheng et al., 2023*), the transcriptome data is insufficient to explain the underlying molecular mechanisms of genetic adaptation in Tibetans. Future single-cell transcriptome analysis and functional validations of the candidate genes are warranted to reveal the responsible cell types and the molecular pathways.

In summary, our study presents a comprehensive analysis of transcriptional profiles and histological differences of placentas of Tibetans and Han migrants. Our data suggest that the male fetuses are more sensitive to high-altitude hypoxia, and the UC layer plays a key role in the observed male-biased transcriptional divergence of the placenta between Tibetans and Han migrants, suggesting its important role in the higher reproductive fitness of Tibetans as the outcome of intense natural selection at high altitude.

## Materials and methods
### Sample collection and processing

We collected 69 healthy full-term placental samples at a hospital in Lhasa (elevation = 3650 m), the capital of Tibet Autonomous Region, China. The detailed information of the investigated samples and statistic results were provided in *Supplementary file 1k and l*. Based on the anatomic structure and the published protocol (*Sood et al., 2006*), each placenta was divided into seven tissue layers, from fetal side to maternal side, including UC, AN, CN, CP, VF, VI, and VM. The UC layer was dissected by cutting ~4 cm from the placental insertion point. The AN and CN layers were obtained by stripping.

We sampled AN from the area ~5 cm from the placental insertion point and CN from the area 3 cm from the placental junction. We obtained the placental parenchyma ~2.5 cm thick at ~3 cm from the placental insertion point. After separation of CP, the remaining part was dissected into triplicates of equal thickness: the maternal layer (including the thin basal plate), the intermediate layer, and the fetal layer. Tissues were snap-frozen in liquid nitrogen and stored at –80°C. Part of the placenta tissue samples were also subject to OCT (optimal cutting temperature compound) embedding for tissue section analysis. All placental samples were obtained and stored within 2 hr of delivery. The information of population ancestry of the sampled subjects was collected by self-claim with no recorded admixture in the past three generations, which was further validated by the genome-sequencing data (*Zheng et al., 2023*).

## RNA extraction and sequencing

Total RNA of the placental samples was extracted using TRIzol reagent, and RNA purity was determined using agarose gel electrophoresis and NanoDrop 2000 (Thermo Scientific). After RNA extraction, we assessed the RNA integrity and purity using agarose gel electrophoresis. The RIN value of the extracted RNA was 7.56±0.71. The extracted RNA was stored at –80°C. RNA library construction and sequencing were conducted through commercial service (Novogene). The Illumina HiSeq 2500 platform was utilized to sequencing with paired-end 150 bp reads. We obtained >6G raw sequencing data for each sample.

## Read alignment and QC

In total, we generated 483 placental RNA-seq data from 69 full-term deliveries of Tibetans (35 placentas) and Han migrants (34 placentas). The QCs including following criteria: (1) filter out low-quality reads; (2) remove the sequence with low-quality ends (the applied threshold was 30); (3) trim reads embracing the joint sequences and the sequences containing the ambiguous base N; (4) remove the transcripts of each sample with reads length <60. After QCs, we obtained a high-quality clean data for subsequent analyses.

The reference human genome (GRCh38 version) and annotation file were downloaded from the *ensemble* database (http://www.ensembl.org/index.html), and the clean reads were then mapped to the reference genome using STAR release 2.6.1a (RRID:SCR_004463) (*Dobin et al., 2013*) with the following default parameters '*--runThreadN 8* --readFilesCommand *gunzip -c -outSAMtype BAM SortedByCoordinate –outSAMmapqUnique 60* --outSAMattrRGline *ID:sample SM:sample PL:ILLUMINA* --outSAMmultNmax *1* --outSJfilterReads *Unique* --outFilterMismatchNmax *2* --alignIntronMin *20* --alignIntronMax *50000* --sjdbOverhang *149* --quantMode *TranscriptomeSAM GeneCounts* --twopassMode *Basic*'. The position information of the sequencing reads on the reference genome and the feature information of each sample were obtained to create bam files. Subsequently, the bam files were fed to the *rsem* software (version: v1.3.3, RRID:SCR_000262) (https://github.com/deweylab/RSEM; *deweylab, 2020*; *Li and Dewey, 2011*) to generate read counts and TPM (transcripts per million reads). In total, 10.6 billion reads were mapped to the annotated regions, and 17,283 genes expressed in all the investigated placenta.

## Determination of fetal and maternal origin of placenta

To trace the maternal-fetal origin of the placenta, we calculated the maternal and fetal ratio by using the maternal genomic data and the fetal genome-array data. An informative SNP is defined as fetal-specific when it is heterozygous in fetus (A/B) and meanwhile is homozygous in mother (A/A) (*Tsang et al., 2017*). Maternal-specific informative SNP definition is opposite. We randomly selected fetal-specific and mother-specific tag SNP (at least five informative SNPs for each sample), respectively. B is the specific allele and A is defined as common allele. For the RNA-seq data of each layer of placental tissue, we calculate allelic ratio (R) according to the following formula:

$$R = \frac{B}{B + A}$$

We calculated five times and got five R values for each sample, and calculated the average value as maternal-specific allelic ratio ($R_m$) and fetal-specific allelic ratio ($R_f$) of the sample.

## Differential expression analysis

Differential expression analysis was performed by R package DESeq2 (*Love et al., 2014*) on the seven layers of placenta. We added two covariates (fetal sex and maternal age) to correct for gene expression: design = ~fetal sex+maternal age. A nominal significance threshold of the adjusted p<0.05 was used to identify DEGs. The Benjamini-Hochberg FDR (false discovery rate) procedure was used in multiple testing correction of p-values. Functional enrichment analysis of DEGs was evaluated using the clusterProfiler (*Yu et al., 2012*); we calculated the FDR-adjusted q-values from the overall p-values to correct for multiple testing. For the gender-separated analysis of differential expression, only maternal age was included in covariates.

## Construction of gene co-expression module

We used R package WGCNA (*Langfelder and Horvath, 2008*) to construct gene expression module for the male-infant placentas and the female-infant placentas separately. The purpose of this analysis is to capture the interactions between gene co-expression modules and reproductive phenotypes. The 11 reproductive phenotypes were included: BW, BPD, FL, GT, PW, PLV, AG, AFMD, AFI, FH, and FUH.

We used *varianceStabilizingTransformation* data from DESeq2 as input expression data. Use the *pickSoftThreshold* function to perform the analysis of network topology and choose a proper soft-thresholding power. Call *blockwiseModules* function to build gene network and identify modules. Then we correlated phenotypic characteristics with summary profile and look for the most significant associations. Considering there are 12 traits, we calculated a Bonferroni threshold (p-value=0.05/number of independent traits) using the correlation matrix of the traits to evaluate the significant modules. The estimated number of independent traits among the 12 investigated traits was 4. Therefore, we used a more stringent significant threshold of p-value=0.0125 (0.05/4) as the final threshold to correct multiple testing brought by multiple traits in the WGCNA.

## Identification of key differential modules and hub genes

The module is defined differential when the p-value<0.05 in population-based WGCNA and p-value<0.05 for phenotype-based WGCNA analysis. We defined a module as key differential modules when a differential module with the most significant p-value, at the same time, with the most significant overlaps with DEGs. Co-expression gene networks were visualized using Cytoscape (v3.7.0, RRID:SCR_003032) (*Shannon et al., 2003*). Functional enrichment analysis of genes in the modules of interest was performed using clusterProfiler.

We defined hub genes with three criteria: (1) gene module correlation >0.2; (2) gene phenotype correlation >0.8; (3) interaction degree ranked in the top 3.

## Simulation analysis

We adopted the permutation approach to evaluate the enrichment of genes overlapped between DEGs and TSNGs. For each permutation, we randomly extracted 192 genes from all the placenta-expressed genes identified from the seven layers (17,284 genes in total), then overlapped them with the three DEG sets (female and male, female only, and male only) and counted the gene number. After 10,000 permutations, we constructed a null distribution for each set, and counted the replicates with equal or more overlapping genes than observed (≥4 for the 'combined' set; ≥9 for the 'male-only' set; ≥0 for the 'female-only' set). We found that the overlaps between DEGs and TSNGs were significantly enriched only in the 'male-only' set (p-value<1e-4, counting 0 time from 10,000 permutations), but not in the 'female-only' set (p-value=1, counting 10,000 time from 10,000 permutations), or 'combined' set (p-value=0.0603, counting 603 time from 10,000 permutations) (*Figure 6—figure supplement 1*). This result suggests that the observed male DEGs are significantly enriched in TSNGs when compared to random sampling.

## Histological analysis

We performed histological analysis for three parts of placenta: UC, VF, and VI, and compared the differences between native Tibetans and Han migrants. In total, 20 individual placenta samples from random selection were analyzed, including 10 from male infants (5 Tibetans vs. 5 Han) and 10 from female infants (5 Tibetans vs. 5 Han). For the UC, we first fixed the UC tissue in 4% paraformaldehyde

48 hr after picked out from –80°C freezer, then cut the middle part of the UC (~0.5 cm long) and fixed overnight. Next, the OCT embedding was performed after sucrose gradient dehydration (15% sucrose dewatering bottling, 30% sucrose dewatering bottling, 4°C), with the UC facing upward. Finally, we performed frozen section (10 µm thickness) and H&E staining. Five parameters of UC were evaluated based on published methods (*Lan et al., 2018*), including area size of UVL, UVW, UAL, UAW, and UAIM. All these parameters were analyzed by ImageJ (*Collins, 2007*). Considering lack of paraffin-embedded material, we placed OCT-embedded tissue in 4% paraformaldehyde for fixation and paraffin embedding. Then, the samples were cut into 0.3 µm thick sections and counterstained to H&E staining. Five fields with equal area per section were randomly selected for counting the numbers of villi and the numbers of syncytial knots, and the average numbers were taken in the Tibetan-Han comparison.

## Acknowledgements

We are grateful to all participants in this study. We would like to acknowledge Chunxia Li, Rongrong Bai, and all other Doctors and Nurses from Obstetrical Department and Delivery Room of Fukang Obstetrics and Gynecology Children's Branch Hospital for their assistance in data collection of the pregnant cohorts.

This study was funded by grants from the National Natural Science Foundation of China (NSFC) (32288101 and 91631306 to BS; 3217040584 and 32000390 to YH; 32070578 and U22A20340 to XQ and 32170629 to HZ), the Youth Innovation Promotion Association of CAS (to YH), the Science and Technology General Program of Yunnan Province (202301AW070010 and 202001AT070110 to YH), the Provincial Key Research, Development and Translational Program of Tibetan Autonomous Region of China (XZ202201ZY0035G to XQ), and the State Key Laboratory of Genetic Resources and Evolution (GREKF22-15 to HZ).

## Additional information

### Funding

| Funder | Grant reference number | Author |
| --- | --- | --- |
| National Natural Science Foundation of China | 32288101 | Bing Su |
| National Natural Science Foundation of China | 91631306 | Bing Su |
| National Natural Science Foundation of China | 3217040584 | Yaoxi He |
| National Natural Science Foundation of China | 32000390 | Yaoxi He |
| National Natural Science Foundation of China | 32070578 | Xuebin Qi |
| National Natural Science Foundation of China | U22A20340 | Xuebin Qi |
| National Natural Science Foundation of China | 32170629 | Hui Zhang |
| Youth Innovation Promotion Association of Chinese Academy of Sciences | | Yaoxi He |
| Science and Technology General Program of Yunnan Province | 202301AW070010 | Yaoxi He |

| Funder | Grant reference number | Author |
| --- | --- | --- |
| Science and Technology General Program of Yunnan Province | 202001AT070110 | Yaoxi He |
| Provincial Key Research, Development and Translational Program of Tibetan Autonomous Region of China | XZ202201ZY0035G | Xuebin Qi |
| State Key Laboratory of Genetic Resources and Evolution | GREKF22-15 | Hui Zhang |

The funders had no role in study design, data collection and interpretation, or the decision to submit the work for publication.

### Author contributions

Tian Yue, Data curation, Formal analysis, Writing - original draft; Yongbo Guo, Formal analysis; Xuebin Qi, Resources, Project administration; Wangshan Zheng, Resources, Formal analysis; Hui Zhang, Bin Wang, Ouzhuluobu, Resources; Kai Liu, Data curation; Bin Zhou, Xuerui Zeng, Validation; Yaoxi He, Formal analysis, Supervision, Writing - original draft, Project administration, Writing - review and editing; Bing Su, Conceptualization, Supervision, Investigation, Writing - original draft, Writing - review and editing

### Author ORCIDs

Tian Yue  http://orcid.org/0009-0003-9259-778X
Yongbo Guo  http://orcid.org/0000-0003-4588-1713
Yaoxi He  http://orcid.org/0000-0003-3324-3239
Bing Su  http://orcid.org/0000-0002-4379-9014

### Ethics

All participants signed the written informed consent. To make sure the local native Tibetans can fully understand the content of the consent, the informed consent was prepared in two language versions (Chinese and Tibetan), and we provided oral interpretation for those who were not able to read. The protocol of this study was reviewed and approved by the Internal Review Board of Kunming Institute of Zoology, Chinese Academy of Sciences (Approval ID: SMKX-20160311-45) and the research scheme is in accordance with the Regulations of the People's Republic of China on the Administration of Human Genetic Resources.

Joint Public Review: https://doi.org/10.7554/eLife.89004.5.sa1
Author response https://doi.org/10.7554/eLife.89004.5.sa2

## Additional files

### Supplementary files
• Supplementary file 1. Excel file including the supplementary tables of this study.
• MDAR checklist

### Data availability

Transcriptome data and phenotypic data generated by this study were deposited at Genome Sequence Archive (GSA). Data can be downloaded under the project ID of PRJCA014064.

The following dataset was generated:

| Author(s) | Year | Dataset title | Dataset URL | Database and Identifier |
|---|---|---|---|---|
| Su B | 2024 | Sex-biased transcriptomic difference in placenta between Tibetans and Han migrants at high altitude | https://ngdc.cncb.ac.cn/bioproject/browse/PRJCA014064 | Genome Sequence Archive, PRJCA014064 |

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
