## [Editor Report · eLife assessment]

This **fundamental** study reports differential expression of key genes in full-term placenta between Tibetans and Han Chinese at high elevations, which are more pronounced in the placenta of male fetus than in female fetus. The gene expression data were collected and analyzed using **solid** and validated methodology, although there is limited support for hypoxia-specific responses due to a lack of low-altitude samples. Several of the placental genes found in this study have been previously reported to show signatures of positive selection in Tibetans, pointing to a potential mechanism of how human populations adapt to high elevation by mitigating the negative effects of low oxygen on fetal growth. The work will be of interest to evolutionary and population geneticists as well as researchers working on human hypoxic response.

---

## [Referee Report · Joint Public Review]

This manuscript by Yue et al. aims to understand the molecular mechanisms underlying the better reproductive outcomes of Tibetans at high altitude by characterizing the transcriptome and histology of full-term placenta of Tibetans and compare them to those Han Chinese at high elevations.

The approach is innovative, and the data collected are valuable for testing hypotheses regarding the contribution of the placenta to better reproductive success of populations that adapted to hypoxia. The authors identified hundreds of differentially expressed genes (DEGs) between Tibetans and Han, including the EPAS1 gene that harbors the strongest signals of genetic adaptation. The authors also found that such differential expression is more prevalent and pronounced in the placentas of male fetuses than those of female fetuses, which is particularly interesting, as it echoes with the more severe reduction in birth weight of male neonates at high elevation observed by the same group of researchers (He et al., 2022).

Comments on latest version:

The revised manuscript has incorporated the suggested changes and weakened conclusions regarding natural selection. Limitations of the study are also clearly stated in the Discussion section.

---

## [Author Response]

The following is the authors’ response to the previous reviews.

**Public Review:**
This manuscript by Yue et al. aims to understand the molecular mechanisms underlying the better reproductive outcomes of Tibetans at high altitude by characterizing the transcriptome and histology of full-term placenta of Tibetans and compare them to those Han Chinese at high elevations.The approach is innovative, and the data collected are valuable for testing hypotheses regarding the contribution of the placenta to better reproductive success of populations that adapted to hypoxia. The authors identified hundreds of differentially expressed genes (DEGs) between Tibetans and Han, including the EPAS1 gene that harbors the strongest signals of genetic adaptation. The authors also found that such differential expression is more prevalent and pronounced in the placentas of male fetuses than those of female fetuses, which is particularly interesting, as it echoes with the more severe reduction in birth weight of male neonates at high elevation observed by the same group of researchers (He et al., 2022).This revised manuscript addressed several concerns raised by reviewers in last round. However, we still find the evidence for natural selection on the identified DEGs--as a group--to be very weak, despite more convincing evidence on a few individual genes, such as EPAS1 and EGLN1.The authors first examined the overlap between DEGs and genes showing signals of positive selection in Tibetans and evaluated the significance of a larger overlap than expected with a permutation analysis. A minor issue related to this analysis is that the p-value is inflated, as the authors are counting permutation replicates with MORE genes in overlap than observed, yet the more appropriate way is counting replicates with EQUAL or MORE overlapping genes. Using the latter method of p-value calculation, the "sex-combined" and "female-only" DEGs will become non-significantly enriched in genes with evidence of selection, and the signal appears to solely come from male-specific DEGs. A thornier issue with this type of enrichment analysis is whether the condition on placental expression is sufficient, as other genomic or transcriptomic features (e.g., expression level, local sequence divergence level) may also confound the analysis.

According to the suggested methods, we counted the replicates with equal or more overlapping genes than observed (≥4 for the “combined” set; ≥9 for the “male-only” set; ≥0 for the “female-only” set). We found that the overlaps between DEGs and TSNGs were significantly enriched only in the “male-only” set (p-value < 1e-4, counting 0 time from 10,000 permutations), but not in the “female-only” set (p-value = 1, counting 10,000 time from 10,000 permutations), or “combined” set (p-value = 0.0603, counting 603 time from 10,000 permutations) (see Table R1 below).

We updated this information in the revised manuscript, including Results, Methods, and Figure S9.

**Author response table 1. sa2table1:** Permutation analysis of the overlapped genes between DEGs and TSNGs.

combined set		male-only set		female-only set	
overlapinggene counts	counts in 10000permutations	overlapinggene counts	counts in 10000permutations	overlapinggene counts	counts in10000permutations
1	4072	1	2418	0	9429
2	3707	2	3572	1	556
3	1619	3	2395	2	16
4	480	4	1124		
5	101	5	373		
6	21	6	92		
7	1	7	25		
		8	2		
		9	0		
Counts >= 4	603	Counts >= 9	0	Counts >= 0	10000
P-value	0.0603	P-value	< 1e-4	P-value	1

The authors next aimed to detect polygenic signals of adaptation of gene expression by applying the PolyGraph method to eQTLs of genes expressed in the placenta (Racimo et al 2018). This approach is ambitious but problematic, as the method is designed for testing evidence of selection on single polygenic traits. The expression levels of different genes should be considered as "different traits" with differential impacts on downstream phenotypic traits (such as birth weight). As a result, the eQTLs of different genes cannot be naively aggregated in the calculation of the polygenic score, unless the authors have a specific, oversimplified hypothesis that the expression increase of all genes with identified eQTL will improve pregnancy outcome and that they are equally important to downstream phenotypes. In general, PolyGraph method is inapplicable to eQTL data, especially those of different genes (but see Colbran et al 2023 Genetics for an example where the polygenic score is used for testing selection on the expression of individual genes).We would recommend removal of these analyses and focus on the discussion of individual genes with more compelling evidence of selection (e.g., EPAS1, EGLN1).

According to the suggestion, we removed these analyses in the revised manuscript.